# RNAi-Mediated Manipulation of Cuticle Coloration Genes in *Lygus hesperus* Knight (Hemiptera: Miridae)

**DOI:** 10.3390/insects13110986

**Published:** 2022-10-27

**Authors:** Colin S. Brent, Chan C. Heu, Roni J. Gross, Baochan Fan, Daniel Langhorst, J. Joe Hull

**Affiliations:** USDA-ARS Arid Land Agricultural Research Center, Maricopa, AZ 85138, USA

**Keywords:** cuticular pigmentation, melanin pathway, *Lygus hesperus*, RNAi

## Abstract

**Simple Summary:**

Insect cuticle coloration results from pigments accumulating in a species-specific pattern. Numerous genes are involved in regulating pigmentation, and their manipulation can be used to create externally visible markers of successful gene editing. To clarify the roles for many of these genes and examine their suitability as markers in *Lygus hesperus* Knight (western tarnished plant bug), we screened existing gene expression data for sequences similar to those with known functions in pigmentation. We identified six genes (*aaNAT*, *black*, *ebony*, *pale*, *tan*, and *yellow*), with two variants for black and found that expression varied for each by developmental stage, adult age, body part, and sex. Silencing the expression of each gene produced varied effects in adults, ranging from the non-detectable (*black 1*, *yellow*), to moderate decreases (*pale*, *tan*) and increases (*black 2*, *ebony*) in darkness, to extremely dark and pervasive pigmentation (*aaNAT*). Based solely on its expression profile and easily observed color changes, *aaNAT* appears to be the best marker for tracking transgenic *L. hesperus*.

**Abstract:**

Cuticle coloration in insects is a consequence of the accumulation of pigments in a species-specific pattern. Numerous genes are involved in regulating the underlying processes of melanization and sclerotization, and their manipulation can be used to create externally visible markers of successful gene editing. To clarify the roles for many of these genes and examine their suitability as phenotypic markers in *Lygus hesperus* Knight (western tarnished plant bug), transcriptomic data were screened for sequences exhibiting homology with the *Drosophila melanogaster* proteins. Complete open reading frames encoding putative homologs for six genes (*aaNAT*, *black*, *ebony*, *pale*, *tan*, and *yellow*) were identified, with two variants for *black*. Sequence and phylogenetic analyses supported preliminary annotations as cuticle pigmentation genes. In accord with observable difference in color patterning, expression varied for each gene by developmental stage, adult age, body part, and sex. Knockdown by injection of dsRNA for each gene produced varied effects in adults, ranging from the non-detectable (*black 1*, *yellow*), to moderate decreases (*pale*, *tan*) and increases (*black 2*, *ebony*) in darkness, to extreme melanization (*aaNAT*). Based solely on its expression profile and highly visible phenotype, *aaNAT* appears to be the best marker for tracking transgenic *L. hesperus*.

## 1. Introduction

*Lygus hesperus* Knight, the western tarnished plant bug, is a significant pest of numerous economically important crops including cotton, alfalfa, and strawberries [1,2,3,4,5]. Recent functional genomics studies [6,7,8,9,10,11,12,13] and the optimization of CRISPR/Cas9 for this species [14], have made it possible to begin developing gene driver systems for use in population control applications. However, tracking successful gene manipulation is difficult when internal traits, such as gametogenesis, are targeted. Co-modification of a highly visible trait can greatly enhance transgenic tracking capabilities. A common tactic is to target genes involved in pigmentation pathways to produce a highly visible external phenotype. This approach has already been used to modify *L. hesperus* eye color [10,14], but the resultant changes can be subtle, and the eyes of this species are relatively small in proportion to their bodies. More efficient screening, particularly of adults, may be realized though modification of cuticular pigmentation. Readily discerned color changes have been successfully induced in several insect species [15,16,17,18,19,20,21], including other hemipterans [22].

Like many insects, *L. hesperus* exhibits variation in body coloration across developmental stages and between sexes [23,24]. During the nymphal stages, both sexes are almost uniformly light green in color with the only substantive accumulation of pigments observed in their reddish-brown eyes and four black spots on the pronotum. After the adult molt, eye color continues to darken for the next 4–7 days until it achieves a dark brown in both sexes [10]. During the same period of maturation, the scutellum develops a bright yellow heart shape contrasted against a black mesoscutum, and the dorsal abdomen becomes uniformly black. The wing membrane transitions from clear to being tinted with black and brown. The legs and antennae become yellowed with additional brown pigments concentrated near the joints, areas of heavy sclerotization. There is also brown banding on the legs that may be a species-specific pattern. The dark spots on the pronotum become extended into dark brown rays and additional dark spots often develop in the corners of this plate. Male coloration on most body parts follows the same patterning as for females but is generally darker. However, unlike females which retain a yellow to green ventral abdomen, due to tinted fat visible under clear cuticle, males develop strips of dark pigment along the lateral margins and a wide black stripe down the center on a yellow to green background [25]. This stark contrast makes it easy to differentiate sexes, so that screening of mature adults is quite rapid.

The pigmentation pathway responsible for insect coloration is well described (Figure 1), but variation exists across species. The genetics of melanization and sclerotization were first delineated in mutant *Drosophila melanogaster* (reviewed in [26]). Melanins are initially derived from tyrosine, which is converted by tyrosine hydroxylase (*pale*) to dihydroxyphenylalanine (DOPA). DOPA is transformed into dopamine (DA) by dopa decarboxylase (DDC). Both DOPA and DA can be converted into their corresponding melanins via several enzymatic reactions through quinone intermediates. This process is promoted by the Yellow binding protein (*yellow*), although the exact function is not yet known [27]. DOPA melanin produces black coloration, while DA melanin produces brown to black. Rather than being used for melanin production, DA can be converted by synthases into colorless or yellow-hued sclerotin pigments. Production of the clear N-acetyldopamine (NADA) sclerotin is catalyzed by aralkylamine N-acetyltransferase (*aaNAT*). Production of the yellow to tan N-β-alanyldopamine (NBAD) sclerotin is promoted by NBAD synthase (*ebony*) and aspartate 1-decarboxylase (ADC or *black*, depending on species), but its reversion is promoted by NBAD hydrolase (*tan*).

Mutations to these key genes have produced a variety of species-specific results. Mutations to *aaNAT* cause increased melanization in *Bombyx mori* [15,28,29], *D. melanogaster* [30], *Oncopeltus fasciatus* [17], *Tribolium castaneum* [31], and *Zootermopsis nevadensis* [32]. In contrast, overexpression or ectopic expression of *aaNAT* causes color reduction in *B. mori*, *D. melanogaster*, and *Harmonia axyridis* [16]. Changes to *black* are associated with much darker pigmentation in *B. mori* [33], *D. melanogaster* [26,34], *Henosepilachna vigintioctopunctata* [21], and *T. castaneum* [35,36]. Similar darkening can occur when *ebony* is impaired, as observed in *B. mori* [37], *Ceratitis capitata* [38], *Drosophila* [19,39], *H. vigintioctopunctata* [21], *O. fasciatus* [17], *Spodoptera litura* [40], *Tenebrio molitor* [41], and *T. castaneum* [42]. Mutation or knockdown of *tan* prevents the conversion of NBAD back to DA for production of DA melanin, resulting in lighter coloration in *Heliconius butterflies* [43], and *D. melanogaster* [19,44,45]. However, knockdown of *tan* had little discernible effect on coloration in *H. vigintioctopunctata* [21]. Interference with *pale* typically reduces melanization, as observed in *Anopheles sinensis* [46], *B. mori* [47], *Manduca sexta* [48], *Spodoptera exigua* [49], and *T. castaneum* [50]. Loss of *yellow* also causes melanin reduction in *Aedes aegypti* [51], *B. mori* [37,52], *Drosophila biarmipes* [27] and *D. melanogaster* [39], *H. vigintioctopunctata* [20,53], *Musca domestica* [54], *Platymeris biguttatus* [55], *S. litura* [18], *T. molitor* [41], and *T. castaneum* [56,57].

Because mutations to these pigmentation-associated genes can result in varied impacts on phenotype when comparing between stages and sexes, and across species, it is necessary to elucidate the role of each gene to determine which might be suitable as a distinctive marker of transgenesis in *L. hesperus*. Based on homology with *Drosophila* protein sequences, *L. hesperus* orthologs of *aaNAT*, *black*, *ebony*, *pale*, *tan*, and *yellow* were identified and characterized via phylogenetic inferences, reverse transcriptase-PCR (RT-PCR) profiling, and RNAi-mediated knockdown.

## 2. Materials and Methods

### 2.1. Experimental Insects

The insects used in this study were obtained from a long-term laboratory colony generated from locally collected *L. hesperus* (USDA-ARS Arid Land Agricultural Research Center, Maricopa, AZ, USA). The stock insects were given ad libitum access to bean (*Phaseolus vulgaris* L.) pods and an artificial diet mix [58] packaged in Parafilm M (Pechiney Plastic Packaging, Chicago, IL, USA) [59]. Food sources were replenished three times per week. Insects were reared in mesh covered containers inside incubators set at 27.0 ± 1.0 °C, 40–60% relative humidity, under a L14:D10 photoperiod. Experimental insects were generated from eggs deposited in Parafilm M agarose packs. Hatchlings were maintained as above.

### 2.2. Bioinformatics

To identify genes involved in *L. hesperus* cuticular pigmentation, previously assembled transcriptomes [7,8] were queried using the online tblastn program (National Center for Biotechnology Information, Bethesda, MD, USA; [60] http://blast.ncbi.nlm.nih.gov/Blast.cgi, accessed on 13 September 2022) with the database search restricted to BioProject accessions PRJNA284294, PRJNA238835, and PRJNA210219. Query sequences consisted of six defined melanin pathway genes (*aaNAT*, *black*, *ebony*, *pale*, *tan*, and *yellow*) from *D. melanogaster* and *O. fasciatus* (accession numbers are listed in Appendix A). Search hits were limited to an e-value ≤ 10^−5^ with the longest and/or lowest e value transcripts re-evaluated via blastx against the nr database. Transcripts identified as encoding putative melanin-associated proteins were conceptually translated and assessed for completeness. Full-length proteins were analyzed for protein structural motifs using Pfam, Gene3D, and Superfamily databases via the HMMER webserver [61]. The identified domain complement was compared to those predicted for the corresponding *D. melanogaster* query sequence.

To determine the relatedness of the two putative *L. hesperus black* homologs, their phylogenetic relationships were evaluated relative to protein sequences (Appendix A) from multiple species across five insect orders (Diptera, Coleoptera, Hemiptera, Hymenoptera, and Lepidoptera). Multiple sequence alignments were generated using MUSCLE 3.8.425 in Geneious Prime (Biomatters Ltd., Auckland, New Zealand; San Francisco, CA, USA). Phylogenetic inferences were constructed in MEGA X [62,63] using the maximum likelihood method with the Le and Gascuel model [64]. Initial tree(s) for the heuristic search were obtained automatically by applying Neighbor-Join and BioNJ algorithms to a matrix of pairwise distances estimated using the JTT model, and then selecting the topology with superior log-likelihood value. A discrete gamma distribution was used to model evolutionary rate differences among sites (5 categories (+G, parameter = 0.6710)) with the analysis incorporating 31 amino acid sequences. All positions with less than 95% site coverage were eliminated, i.e., fewer than 5% alignment gaps, missing data, and ambiguous bases were allowed at any position (partial deletion option) with the final dataset consisting of 477 positions. The same dataset was also evaluated using the neighbor-joining method [65] with evolutionary distances computed using the JTT matrix-based method.

### 2.3. PCR

To verify the predicted *L. hesperus* melanin-associated transcripts, open reading frames (ORFs) were PCR amplified and sequenced. Total RNAs were isolated from individual male or female adults 7 days post-eclosion using TRI Reagent (Life Technologies, Carlsbad, CA, USA) and an RNeasy Mini QIAcube kit (Qiagen, Germantown, MD, USA). First-strand cDNAs were generated with Superscript III reverse transcriptase (Life Technologies) using 500 ng DNase I-treated total RNAs and custom made random pentadecamers (IDT, San Diego, CA, USA). Transcripts of interest were amplified with Sapphire Amp Fast PCR Master Mix (Takara Bio USA Inc., Mountain View, CA, USA) in 20-μL volumes using primers (Appendix A) designed to span each of the predicted ORFs. Thermocycler conditions and cDNA inputs varied for each of the targets. Products for *aaNAT*, *black 1*, *ebony*, and *tan* were generated from 1 μL cDNA using a cycling program that consisted of 95 °C for 2 min followed by 37 cycles at 95 °C for 20 s, 54 °C for 20 s, 72 °C for 1.5 min, and a final extension at 72 °C for 5 min. *Black 2* and *pale* were likewise amplified using 1 μL cDNA but utilized an annealing temperature of 57 °C. *Yellow* was amplified using 2 μL cDNA and an annealing temperature of 57.5 °C. The resulting products were separated on 1.5% agarose gels using a Tris/acetate/EDTA buffer system, visualized with SYBR Safe (Life Technologies) on an Azure 200 Gel Imaging Workstation (Azure Biosystems, Dublin, CA, USA), and then processed using Adobe Photoshop v21.2.12 (Adobe Systems Inc., San Jose, CA, USA). Reaction products were sub-cloned into pCR2.1-TOPO TA (Life Technologies) and sequenced at the Arizona State University DNA Core Laboratory (Tempe, AZ, USA). Consensus sequences have been deposited with GenBank under accession numbers: OP419589 (*Lh aaNAT*), OP419590 (*Lh black 1*), OP419591 (*Lh black 2*), OP419592 (*Lh ebony*), OP419593 (*Lh tan*), OP419594 (*Lh pale*), and OP419595 (*Lh yellow*).

For expression profiling analyses, first-strand cDNAs were generated as above from two biological replicates of pooled eggs (20 per replicate), pooled sets for each of the five nymphal stages (5 nymphs per pooled stage), mixed sex adults (0, 7, and 20 days post-eclosion; 1 adult of each sex per day), and 7-day-old, sexed adult body segments (5 pooled heads, 3 pooled thoraxes, and 3 pooled abdomens). Fragments (~500-bp) of the *L. hesperus* melanin pathway transcripts and actin (GBHO01044314.1) were amplified as above using Sapphire Amp Fast PCR Master Mix from each biological replicate with gene specific primers (Appendix A). Thermocycler conditions consisted of 95 °C for 2 min followed by 35 cycles at 95 °C for 20 s, 56 °C for 20 s, and 72 °C for 30 s, and a final extension at 72 °C for 5 min. PCR products were separated on 1.5% agarose gels and visualized as before. A subset of the reactions was subcloned into pCR2.1-TOPO TA and sequence validated.

### 2.4. RNAi-Mediated Knockdown

The complete 687-bp ORF of *venus* (a yellow fluorescent protein variant) and ~500-bp fragments of the *L. hesperus* melanin pathway transcripts were amplified from validated plasmid DNAs using primers (Appendix A) containing a 5′ T7 promoter sequence (TAATACGACTCACTATAGGGAGA) and Sapphire Amp Fast PCR Master Mix. Thermocycler conditions consisted of 95 °C for 2 min followed by 35 cycles at 95 °C for 20 s, 61 °C for 20 s, and 72 °C for 30 s, and a final extension at 72 °C for 5 min. Double-stranded RNAs (dsRNAs) were generated and purified from the T7 PCR products with a MEGAscript RNAi kit (Life Technologies) and then diluted to 1 µg/µL in MEGAscript RNAi kit elution buffer (EB).

Following cold immobilization on ice for 5 min, dsRNAs (250 nL; 1 µg/µL) were injected into 1-day-old fifth instar nymphs. dsRNA was delivered between the fifth and seventh abdominal tergites using a Nanoject III programmable nanoliter injector (Drummond Scientific Company, Broomall, PA, USA) fitted with a 5-µL disposable soda lime glass pipet needle (ThermoFisher, Kimble Chase, Pittsburgh, PA, USA) as described previously [10]. Because nymphs injected with dsRNAs targeting the putative pale homolog exhibited increased mortality during eclosion, evaluation of the function of pale in pigmentation was conducted by injecting (500 nL; 1 µg/µL) newly eclosed adults of each sex. *L. hesperus* that failed to recover within 10 min post-injection were discarded, all others were maintained under normal rearing conditions until 7 days post-injection as described previously [10]. Treatment groups (40 nymphs of each sex per dsRNA target) were replicated at least three times.

To confirm target transcript knockdown, subsets of each injected group were assayed by RT-PCR as above using primer pairs that do not anneal within the dsRNA target region. Unless otherwise stated, thermocycler conditions consisted of 95 °C for 2 min followed by 35 cycles at 95 °C for 20 s, 56 °C for 20 s, and 72 °C for 30 s, and a final extension at 72 °C for 5 min. Knockdown confirmation of *aaNAT* and *tan* transcripts was assessed using 37 cycles and a 54 °C annealing temperature, whereas for *yellow* and *pale* the thermocycler conditions were modified to 37 cycles at 56 °C and 33 cycles at 53 °C, respectively. PCR products were electrophoresed and visualized as before with gel images processed in Adobe Photoshop. Transcript knockdown was confirmed in at least five replicates.

Phenotypic effects on cuticular coloration were compared across treatment groups using images captured on a Nikon SMZ18 stereomicroscope equipped with a D5-Ri2 camera (Nikon Instruments Inc., Melville, NY, USA). Images were obtained of dorsal, ventral, and lateral whole bodies of each sex, as well as close-ups of the compound eye and dissected pronotum, antennae, hindleg, and forewing. To minimize potential variation, images were taken under consistent magnifications, light conditions, aperture, exposure time, and white balance.

## 3. Results

### 3.1. Identification and Bioinformatic Analyses

Potential components of the *L. hesperus* melanin pathway were identified from publicly available transcriptomic datasets based on sequence homology with genes (*aaNAT*, *black*, *ebony*, *pale*, *tan*, and *yellow*) characterized in *D. melanogaster* and *O. fasciatus* (Appendix A). Reciprocal BLAST searches of the longest and/or lowest e value *L. hesperus* hits against the NCBI nr database support the initial annotations (Appendix A). The longest transcripts identified for all the *L. hesperus* homologs are predicted to encode complete ORFs with protein domains and motifs consistent with those present in the *D. melanogaster* proteins (Appendix A). Based on multiple sequence alignments (Appendix A), the *L. hesperus* melanin pathway proteins exhibit 60–92% similarity with homologs from other species (*D. melanogaster*, *Bombyx mori*, *O. fasciatus*, and *Halyomorpha halys*). This degree of similarity is comparable to that exhibited across species.

Two transcripts were identified as potential *black* homologs. Both contain comparable protein domains but, unlike the *D. melanogaster* protein, which has been annotated as an aspartate 1-decarboxylase, the *L. hesperus* transcripts had highest similarity with sequences annotated as cysteine sulfinic acid decarboxylases (Appendix A). Homologs of *D. melanogaster black*, however, frequently have a different enzymatic annotation. Although initially named based on its phenotypic effects [34], the *D. melanogaster* protein, as well as a number of *black* homologs, function as an aspartate 1-decarboxylase [28,36,41]. The mosquito homolog, which uses cysteine sulfinic acid as a substrate, was correspondingly annotated as a cysteine sulfinic acid decarboxylase [66] as was the *T. castaneum* enzyme [67]. Consequently, the top hits for the *L. hesperus black* homologs are consistent with the current nomenclature.

To assess the phylogenetic relatedness of the *L. hesperus* Black proteins, we constructed maximum likelihood inferences with homologs from representative species across five insect orders (Diptera, Coleoptera, Hymenoptera, Hemiptera, and Lepidoptera). Consistent with other groups [22], we found that the mosquito Black homologs sorted with a coleopteran clade rather than other members of the Diptera (Figure 2). Further, while most of the proteins segregated based on insect order, *L. hesperus* Black 1 and Black 2 (as well as some heteropteran homologs), exhibited a different evolutionary lineage than the canonical Black. A neighbor-joining approach yielded a tree with similar phylogenetic topology (Appendix A), suggesting that the observed clade architectures likely reflect evolutionary history rather than a methodological artifact. Previous phylogenetic analysis of putative black homologs likewise placed cysteine sulfinic acid decarboxylases in a separate clade [68].

### 3.2. RT-PCR Cloning and Expression Profiling

Primers designed to amplify complete ORFs for the *L. hesperus* melanin pathway genes generated PCR products of the expected sizes (Figure 3). Sequence analysis of the consensus clones indicated that all but the *yellow* and *ebony* cDNA products exhibited >99% nucleotide identity with the transcriptomic data. The *yellow* product was 98.3% identical (1298/1320 nt) and had 11 nonsynonymous changes that likely reflect allelic variation in the *L. hesperus* colony. The ebony product was 96.5% identical (2645/2637 nt) with 44 nonsynonymous changes most of which map to the last third of the coding sequence (Appendix A). This differentiation could indicate that the cloned product and the transcriptomic sequence arose from alternative exon usage like that reported for dopa decarboxylase and laccase 2 [36,69].

To provide insights into potential functional roles, we assessed transcriptional expression of the *L. hesperus* melanin pathway genes in eggs, across nymphal development, and in adults at varying points of maturation. Transcripts for all the pathway genes were amplified to varying degrees from all stages of development with the most pronounced expression typically observed early in nymphal development and newly emerged adults (Figure 4A). Primers for the *black 1* transcript unexpectedly yielded a second band that encoded a black 1 variant with a 189-nt insert at position 929 that introduces a premature stop codon that would yield a 321 aa protein rather than the 484 aa protein encoded on the full-length transcript. As with many of the other melanin pathway genes, expression of the *black 1* variant was highest in eggs and newly emerged adults.

We also examined expression in the body segments of reproductively mature male and female adults at 7 days post-eclosion (Figure 4B). None of the genes examined were restricted to a particular body segment nor did they exhibit sex-specific expression. The presence of more robust *black 2* amplimers derived from head cDNAs for both sexes relative to the other body segments, however, is suggestive of preferential expression. Similarly, abdominal expression of yellow was more pronounced in females than males, whereas head expression was the opposite. As with the developmental profile, the *black 1* variant was inconsistently amplified across segments in both sexes.

### 3.3. RNAi-Mediated Knockdown

To determine the functional role played by the putative melanin synthetic pathway genes of *L. hesperus*, we injected dsRNAs corresponding to ~500-bp fragments of the respective coding sequences. This treatment significantly reduced transcript levels in adults at seven days post-injection relative to controls for all the genes targeted (Figure 5). Typically, cuticle pigmentation in wild-type *L. hesperus* is completed in the 5–7 days following eclosion [25]. The pigment patterning is nearly identical between sexes, but relative to a female (Figure 6A), the cuticle of a male (Figure 6B) is substantially darker. The only exception to pigment distribution is on the ventral abdomen, which is unpigmented in females and covered by a large expanse of black pigment in males. As expected, the knockdown of the *Venus* control produced no changes from the wild phenotype. In contrast, the *aaNAT* knockdown caused both males and females to become almost uniformly black on every surface (Figure 6 and Figure 7). Color continued to darken as the adults aged. This was the most pronounced effect of all the genes tested, but many of the other dsRNAs also had discernable effects. The exceptions to this were *black 1* and *yellow*, which produced no change in external phenotype. Darker pigmentation in both females and males was caused by knockdown of *black 2* and *ebony*. The distribution pattern remained the same, but pigmented regions became more pronounced and blacker. The effect of *black 2* was greater than that of *ebony*. Silencing *pale* caused a pronounced loss of pigment on the typically colored sections of the abdomen and thorax in both females and males, with a more evident effect on the latter group (Figure 6), however it did not have a notable impact on eye, antennae, pronotum or legs (Figure 7). Knockdown of *tan* had a negligible impact on the main body (Figure 6) but caused the typical leg banding and wing spotting to become diminished in both sexes (Figure 7). Collectively, these results support the putative role that these genes play in melanin synthesis and cuticle pigmentation, but also suggest region-specific roles impacting color patterning.

## 4. Discussion

With the intent of identifying an obvious external marker for tracking transgenesis in *L. hesperus*, we examined the function of the putative primary genes in the insect pigmentation pathway using RNAi. Analysis of previously generated transcriptomes for this species [6,8] yielded homologs for *aaNAT*, *black*, *ebony*, *pale*, *tan* and *yellow*. All of these genes were observed to be expressed during the period of adult pigment deposition, and injection of dsRNA greatly reduced expression for each respective gene. RNAi did produce evident effects on pigmentation phenotypes, although the specific effect and extent varied by gene.

Silencing the genes involved in the synthesis of the yellow- and tan-colored NBAD sclerotins (Figure 1; *black*, *ebony*, *tan*) had a discernable impact on pigmentation. RNAi of *black* and *ebony*, which are sequentially responsible for converting L-aspartic acid to NBAD, resulted in regions of darker pigmentation, particularly evident where shades of yellow might normally occur. This is likely a result of excess dopamine that was not incorporated into NBAD being utilized to produce DA melanin. It should be noted that RNAi of only *black 2* modified the external phenotype, whereas silencing *black 1* had no discernable effect. Unlike *black 2*, expression of *black 1* varied across replicates, and co-expression of a nonfunctional splice variant was frequently observed. Furthermore, while *black 2* was expressed only during the early stages of adulthood when primary pigmentation was occurring, *black 1* was observed well past when pigmentation had ceased. The phenotypic results and sequence similarity suggest that *black 2* is likely acting as cysteine sulfinic acid decarboxylase, as observed in other species [66,67], driving the synthesis of NBAD. The enzymatic function of *black 1* apparently differs and may not be involved in pigmentation synthesis.

The other gene typically involved in the NBAD sclerotin pathway is *tan*, which produces a hydrolase involved in the conversion of NBAD to DA, making it available for production of DA melanin. The Tan protein appears to play the same role in *L. hesperus*; silencing the gene resulted in lighter coloration, as has been observed in butterflies [43], 2011, and fruit flies [19,44,45]. However, this change was principally localized to the legs and wings, where the intensity of typically dark spots and bands were diminished.

While the RNAi of genes in the NBAD sclerotin pathway does produce discernable effects, the changes are not very pronounced and differentiation from wildtype requires close scrutiny under a microscope, making the phenotypes less than ideal as markers of transgenesis. By comparison, knockdown of *aaNAT*, which is involved in converting DA to NADA, a precursor for the clear NADA sclerotin (Figure 1), produces a change that is readily evident to an unaided eye. Almost every surface on both females and males becomes very darkly melanized, turning the insect black. This phenotype has been observed in other species [15,17,28,30,31,32,33], supporting the putative function of *aaNAT* in *L. hesperus*. RNAi of *Lh aaNAT* inhibits the production of clear sclerotin making more dopamine available for melanization. While this phenotype may be an excellent marker, there are possible side effects which may adversely impact physiology and behavior. In fact, *aaNAT* has been proposed as a target for insecticide development due to the potential severity of these effects [70,71]. The first issue is the potential reduction of NADA sclerotins, which could negatively affect structural integrity and morphology [31,36,72]. While *L. hesperus* may be able to compensate by producing more NBAD sclerotin, this might not produce sufficient rigidity at the parts of the cuticle most in need of reinforcement. Further, aaNAT is responsible for catabolizing DA, which, in addition to being a pigment precursor, is a key neurotransmitter regulating many aspects of insect behavior and physiology [73]. In *L. hesperus*, DA has already been shown to play a critical role in reproduction [74]. Additional study will be needed to ensure that manipulation of *aaNAT* does not impair the functionality of the insects.

The two remaining genes studied, *pale* and *yellow*, are involved in the conversion of tyrosine to melanins. The enzymatic role of *pale* is discreet, acting as a tyrosine hydroxylase to produce DOPA. In other species, interference with *pale* reduces the melanin in the cuticle, causing it to become lighter [47,48,49,50]. In *L. hesperus*, *pale* appears to have the same function, with RNAi producing a loss of pigment in males and females, albeit almost exclusively on the main body and not the antennae, legs and wings. Because the wings cover the area most impacted following *pale* knockdown (i.e., near complete melanization of the dorsal abdomen), targeting this gene would not be ideal for rapid screening as it would require that each insect be handled for inspection.

Interference with *yellow* also typically reduces cuticle melanin [17,20,27,37,39,41,51,52,54,55,56], however, RNAi had no discernable effect on the *L. hesperus* cuticle. There are several possible reasons why no change occurred. First, the search of our transcriptomes search revealed at least eight non-overlapping *yellow*-like transcripts. Such diversity is common and appears to represent recent rapid diversification (reviewed in [20]). Because the function of the numerous isoforms can vary, the single version of *yellow* targeted in this study may not be directly involved in pigmentation. Second, even if this particular *yellow* gene was involved in melanization, its contribution could be highly localized; while loss of yellow causes a body-wide melanization reduction in *Drosophila* [39], knockdown of *yellow* only impacts hindwing pigmentation in *Tribolium* [56], and predominantly abdominal pigmentation in *O. fasciatus* [17]. There is some variation in the tissue expression of *Lh yellow* (Figure 4), but not enough to suggest that it was having an effect that we might have missed. The pigmentation function of yellow can also be stage-specific, as seen in *H. vigintioctopunctata* [20], and may be more operational during nymph development than the period of adult maturation that was examined here. As observed in *T. castaneum* [56], *Vanessa cardui* [75], and *H. vigintioctopunctata* [20], *Lh yellow* expression varied across development, with nothing observable at 7 or 20 days post-eclosion, suggesting that the function of this gene does have a temporal element. A more detailed tissue- and time-specific study of the expression of this yellow and the other *L. hesperus yellow*-like transcripts would be needed to fully understand the roles played by each, but that is well outside of the scope of our endeavor to identify a prominent marker for transgenesis.

In summary, the pigmentation pathway genes of *L. hesperus* appear to share many of the same functional attributes as those observed in other species. There is evidence of region- and sex-specific patterning being generated by differential expression of individual genes, and of multiple genes working together to generate the final color phenotype. While RNAi of most of the genes examined had discernable effects on pigmentation patterns, only knockdown of *aaNAT* produced a phenotype that was sufficiently distinctive to allow rapid screening of both females and male for successful transformation. Future efforts will focus on developing CRISPR to induce this mutation in embryos, and on ensuring the resultant mutant phenotype does not cause any negative effects on survival and reproduction.

## Figures and Tables

**Figure 1 insects-13-00986-f001:**
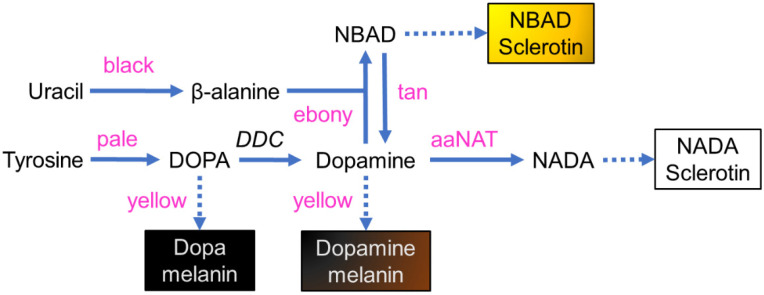
Model of melanin and sclerotin biosynthesis in insect cuticles. Tyrosine is converted by tyrosine hydroxylase (pale) to DOPA, which in turn can be transformed by DDC into dopamine. DOPA and dopamine can be converted into brown and black melanins through several intermediate steps that are promoted by yellow. Dopamine also can be converted by synthases into colorless or yellow sclerotin pigments. Production of clear NADA sclerotins is catalyzed by aaNAT. Production of the yellow to tan NBAD sclerotin is promoted by ebony and black, but its reversion is promoted by tan. Adapted from [17,26].

**Figure 2 insects-13-00986-f002:**
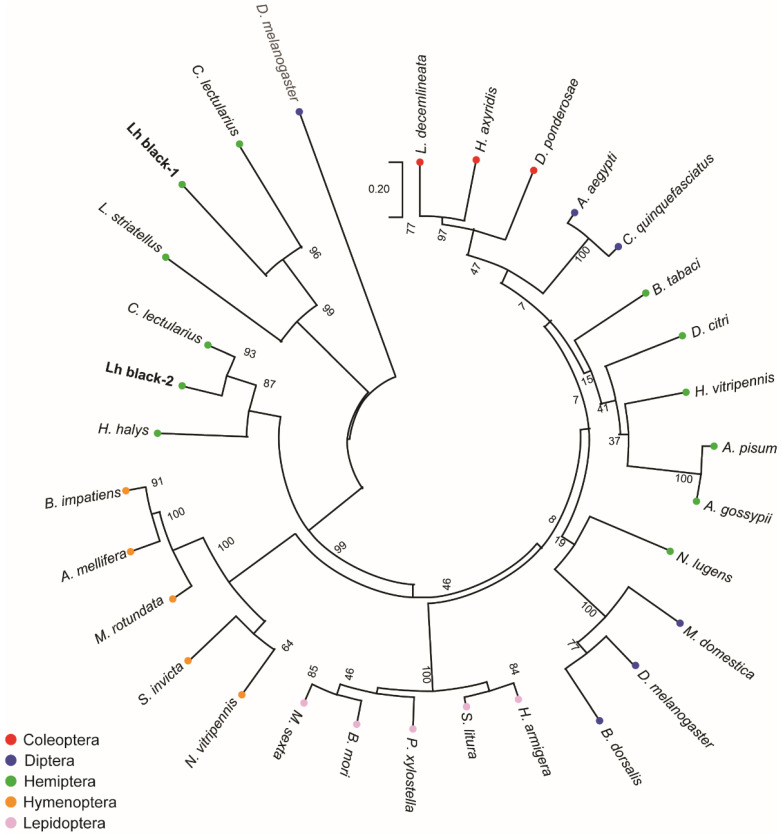
Phylogenetic analysis of putative black homologs in *L. hesperus* and other insects. The tree with the highest log likelihood (−14,891.07) is shown. The tree was rooted to the *Drosophila melanogaster* glutamic acid decarboxylase 1 and is drawn to scale, with branch lengths measured in the number of substitutions per site. Insect orders have been color coded: Coleoptera—red; Diptera—blue; Hemiptera—green; Hymenoptera—orange; and Lepidoptera—pink. Numbers at the nodes indicate bootstrap support across 1000 replicates. Accession numbers for the sequences used are listed in Appendix A.

**Figure 3 insects-13-00986-f003:**
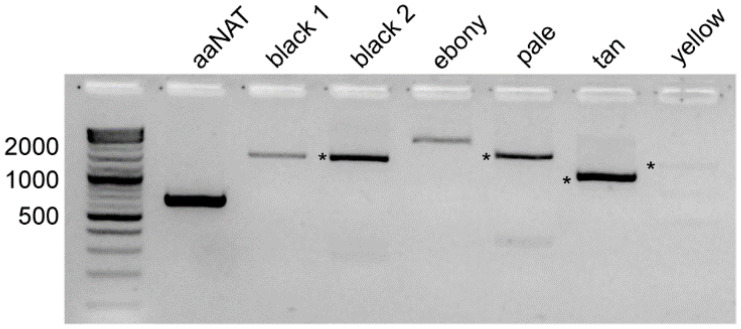
RT-PCR amplification of melanin pathway transcripts. Products were generated from mature adult body cDNAs using primers designed to amplify complete open reading frames. Expected sizes are: *aaNAT*—669 bp; *black 1*—1455 bp; *black 2*—1542 bp; *ebony*—2343 bp; *pale*—1680 bp; *tan*—1131 bp; and *yellow*—1320 bp. In lanes with multiple bands, products of the expected sizes are marked with an asterisk. All products were subcloned and sequenced.

**Figure 4 insects-13-00986-f004:**
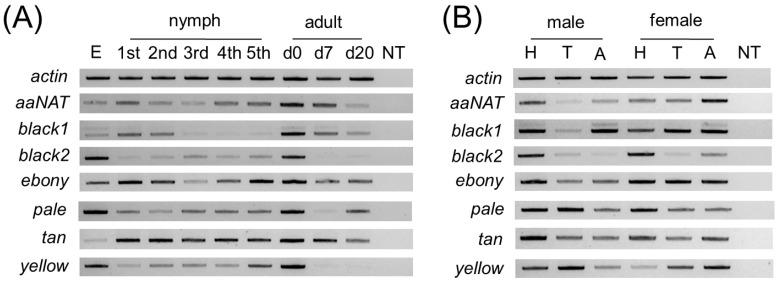
Expression profile of *Lygus hesperus* cuticular pigment-associated transcripts. (**A**) Developmental expression of transcripts from egg (E) through nymphal development (1st–5th), and at varying points in adult maturation including adults at emergence (d0), and at 7 (d7), and 20 days (d20) post-eclosion. (**B**) Limited sex-specific tissue profile using head (H), thoracic (T), and abdominal (**A**) segments of reproductively mature adults (i.e., 7 day-old). Amplimers are ~500-bp fragments of the transcripts of interest. Gels shown are representative of results obtained across three replicates.

**Figure 5 insects-13-00986-f005:**
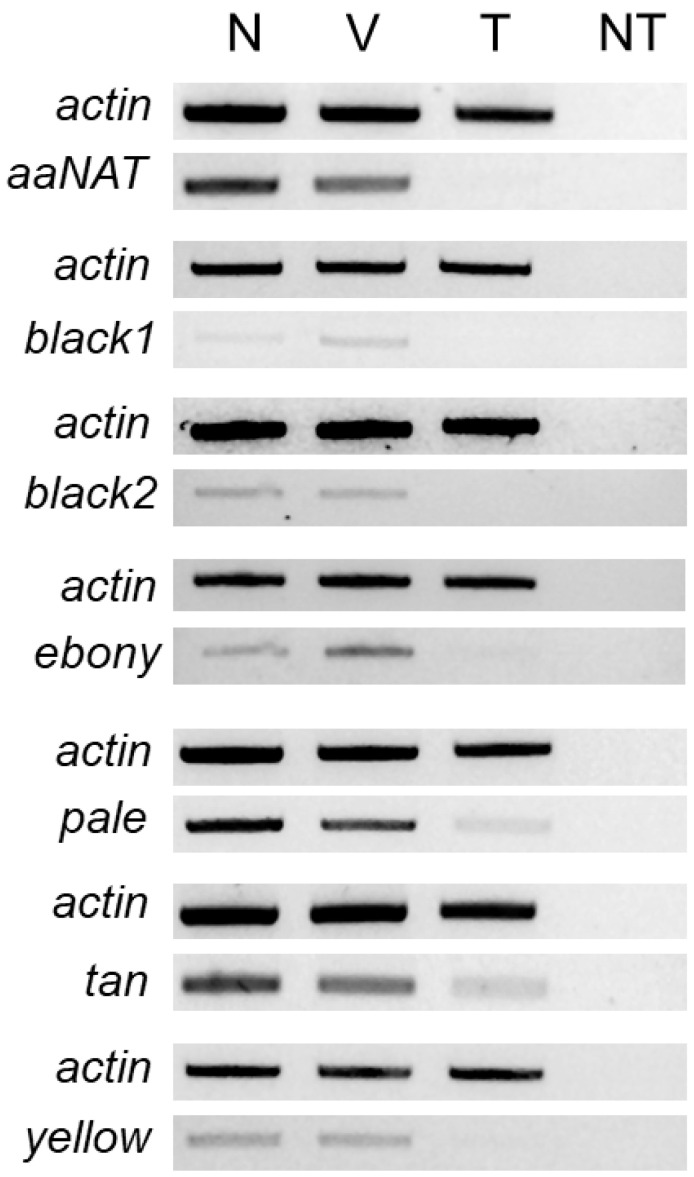
RT-PCR confirmation of target transcript knockdown by dsRNA-mediated RNAi. Products correspond to ~500-bp fragments of the transcripts of interest amplified from single adults at 7 days post-injection. Gels shown are representative of results obtained across five replicates. Abbreviations are: N (noninjected); V (*venus* dsRNA injected); T (target dsRNA injected); NT (no template).

**Figure 6 insects-13-00986-f006:**
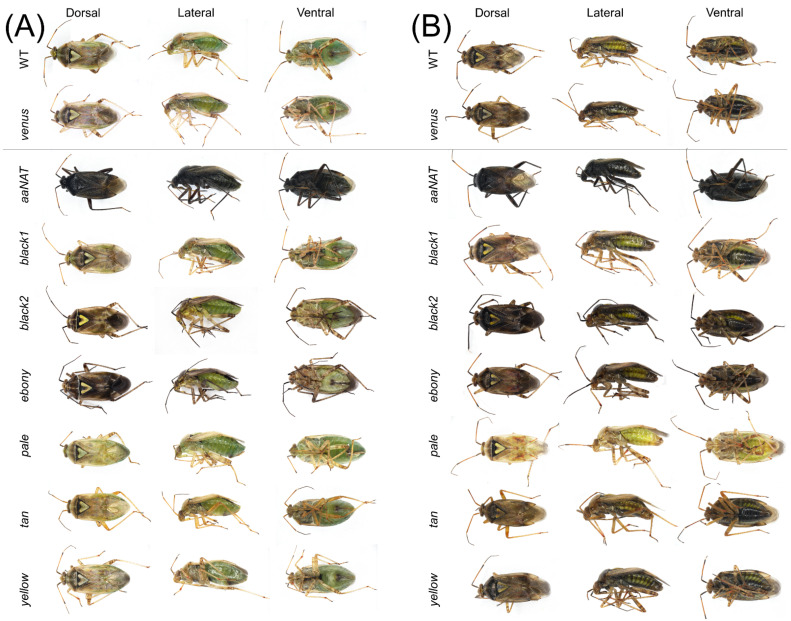
Effects of RNA interference-mediated knockdown on *Lygus hesperus* cuticle coloration. Shown are full-body images (dorsal, lateral and ventral views) of (**A**) female and (**B**) male *L. hesperus* adults at 7 days post-eclosion that were either untreated wildtype (WT) or injected as 5th instars with dsRNA of the control gene venus, or the indicated cuticle pigment-associated gene. Images are representative of three biological replicates consisting of 40 bugs of each sex.

**Figure 7 insects-13-00986-f007:**
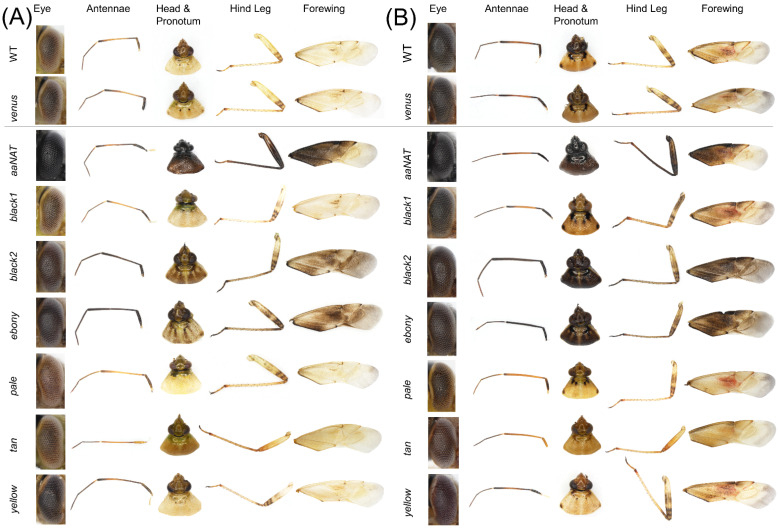
Effects of RNA interference-mediated knockdown on *Lygus hesperus* cuticle coloration of individual body parts. Shown are the eyes, antennae, pronotum and head, rear right leg, and right forewing of (**A**) female and (**B**) male *L. hesperus* adults at 7 days post-eclosion that were either untreated wildtype (WT) or injected as fifth instars with dsRNA of the control gene *venus*, or the indicated cuticle pigment-associated gene. Images are representative of three biological replicates consisting of 40 bugs of each sex.

## Data Availability

On reasonable request, derived data supporting the findings of this study are available from the corresponding author.

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
