# Peer review of "RNAi-Mediated Manipulation of Cuticle Coloration Genes in Lygus hesperus Knight (Hemiptera: Miridae)"

_insects, 2022, doi:10.3390/insects13110986_

Round 1
Reviewer 1 Report
Brent and colleagues found that knockdown of aaNAT by dsRNA showed extremely dark pigmentation in both male and female L. hesperus. They proposed that aaNAT could be a marker for tracking transgenic L. hesperus. Overall, this is a really interesting study that would be of interest for researchers working on insect cuticle coloration and transgenic insects.
Below are several comments and criticisms that need to be addressed before the manuscript can be considered for publication.
Line 202, for expression profiling analyses, how many eggs, nymphs or adults did the authors use in each biological replicate?
Line 366-367, the RT-PCR products were amplified from the transcripts of single adult at 7 days post-injection. It is more reliable to mix several insects to confirm the knockdown effect of dsRNA although one adult may be enough to extract RNA for RT-PCR analysis.
Author Response
Reviewer: Line 202, for expression profiling analyses, how many eggs, nymphs or adults did the authors use in each biological replicate?
Response: This information was accidentally omitted and is now found in lines 184-187.
Reviewer: Line 366-367, the RT-PCR products were amplified from the transcripts of single adult at 7 days post-injection. It is more reliable to mix several insects to confirm the knockdown effect of dsRNA although one adult may be enough to extract RNA for RT-PCR analysis.
Response: The RT-PCR knockdown results shown for each gene target are representative of confirmed knockdown in at least five replicates (see lines 225-226). Although pooling samples may be useful in some instances, in our experience we have found that doing so can skew the results due to an outlier individual included in the mix (e.g. a specimen that for some reason had little knockdown due to poor injection). For this reason, we typically assess knockdown multiple times at the individual level rather than using pooled samples. When we ran tissue samples rather than whole bodies, we did pool to amplify the signal (see lines 183-187).
Reviewer 2 Report
The manuscript by Brent et al presents the identification and functional analysis of cuticle pigmentation genes in Lygus hesperus, a plant bug, as part of their efforts to define easily tractable phenotypes for transgenesis. The writing is fluid, with objectives well stated and data clearly presented. This is an important contribution to the field, as it provides information to build the toolbox required for gene edition and biological control strategies directed to this agricultural pest. I have only a few observations, listed below.
Results
1) Gene identification, expression analysis and knockdown effectiveness are appropriately established, although, for the knockdowns, some RT-PCR data would gain from showing stronger bands, particularly for aaNAT, pale and tan, where control PCR bands are very light.
2) Further, some improvement in the quality of Figure 6 would greatly help the reader to visualize the effects on cuticle pigmentation resulting from the different knockdowns. While general pigmentation differences may be perceived, subtle effects are not. Even though the authors provide more detailed analysis of effects on body structures in Fig 7, the image quality still requires improvement. Maybe the quality of Figs 6 and 7 was lost while mounting the pdf and would be of higher quality in an online version. If not, I suggest that authors upload high quality images as part of the final version of the manuscript, as visible phenotypes are central data displayed.
Discussion
3) In relation to eye color, displayed in Fig. 7. How do the authors explain the eye color phenotypes, since they are altering the expression of cuticle pigmentation genes? In general the pigments in ommatidea (primary and secondary pigment cells) are generated through tryptophan and pteridin pathways, and mobilized by ABC transporters. Is there a colored cuticle that covers the eye? Wouldn't this alter the insect's vision?
4) The effect of aaNAT knockdown is striking, as shown for P.bigutattus (Zhang et al, Int J Molec Sci, 2019) and R. prolixus (Berni et al, Genetics, 2022), but not for O. fasciatus (Liu et al, Genetics, 2016). Thus, the authors suggest that aaNAT could be a good candidate as marker for transgenesis, since the phenotype is easily detected. However, they point out that aaNAT knockdown could have adverse effects that would hamper its use, such as loss of cuticle integrity (as shown for R. prolixus), although insects could compensate by producing more NBAD sclerotin. This is a very interesting point, particularly considering that the NBAD branch appears to vary in importance to cuticle pigmentation among hemiptera. For instance, in R. prolixus and P.bigutattus, that display a strong aaNAT knockdown phenotype, tan and ebony exert no effect on cuticle pigmentation. Contrarily, aaNAT knockdown displays only subtle phenotypes in O. fasciatus, which maintains ebony and tan function in pigmentation. How do the authors interpret their results in light of this variable role of the NBAD versus NADA branches of the melanization pathway among Hemiptera?
Author Response
Reviewer: Gene identification, expression analysis and knockdown effectiveness are appropriately established, although, for the knockdowns, some RT-PCR data would gain from showing stronger bands, particularly for aaNAT, pale and tan, where control PCR bands are very light.
Response: We have rerun the samples and have much stronger banding for the many of the genes. Some genes are expressed in quite low amounts (i.e. black1) so there was little change in the images. The new thermocycler conditions are described in lines 219-223.
Reviewer: Further, some improvement in the quality of Figure 6 would greatly help the reader to visualize the effects on cuticle pigmentation resulting from the different knockdowns. While general pigmentation differences may be perceived, subtle effects are not. Even though the authors provide more detailed analysis of effects on body structures in Fig 7, the image quality still requires improvement. Maybe the quality of Figs 6 and 7 was lost while mounting the pdf and would be of higher quality in an online version. If not, I suggest that authors upload high quality images as part of the final version of the manuscript, as visible phenotypes are central data displayed.
Response: Lower quality figures were embedded in the original manuscript submission. Final versions of figures 6 and 7 are of much higher quality and can be enlarged online to provide adequate discernment of the subtle phenotypic differences induced by RNAi.
Reviewer: In relation to eye color, displayed in Fig. 7. How do the authors explain the eye color phenotypes, since they are altering the expression of cuticle pigmentation genes? In general the pigments in ommatidea (primary and secondary pigment cells) are generated through tryptophan and pteridin pathways, and mobilized by ABC transporters. Is there a colored cuticle that covers the eye? Wouldn't this alter the insect's vision?
Response: We have previously used RNAi to research the development of eye color in L. hesperus and did confirm the roles played by various genes implicated in the pteridine and ommachrome pathways. Eye color changes have been observed in other species in which cuticle pigment genes were targeted. With RNAi of LhaaNAT, it appears that even areas in which would normally have clear sclerotization, such as the lenses of ommatidia, become melanized. The mechanism by which this occurs is not currently known and is well beyond the scope of the research presented here. However, these heavily pigmented Lygus do appear visually capable, retaining photoresponsiveness and a capacity to orient themselves during flight. We would rather not detail those findings in this paper as that work is being included in a forthcoming manuscript.
Reviewer: The effect of aaNAT knockdown is striking, as shown for P.bigutattus (Zhang et al, Int J Molec Sci, 2019) and R. prolixus (Berni et al, Genetics, 2022), but not for O. fasciatus (Liu et al, Genetics, 2016). Thus, the authors suggest that aaNAT could be a good candidate as marker for transgenesis, since the phenotype is easily detected. However, they point out that aaNAT knockdown could have adverse effects that would hamper its use, such as loss of cuticle integrity (as shown for R. prolixus), although insects could compensate by producing more NBAD sclerotin. This is a very interesting point, particularly considering that the NBAD branch appears to vary in importance to cuticle pigmentation among hemiptera. For instance, in R. prolixus and P.bigutattus, that display a strong aaNAT knockdown phenotype, tan and ebony exert no effect on cuticle pigmentation. Contrarily, aaNAT knockdown displays only subtle phenotypes in O. fasciatus, which maintains ebony and tan function in pigmentation. How do the authors interpret their results in light of this variable role of the NBAD versus NADA branches of the melanization pathway among Hemiptera?
Response: First, we are grateful for the mention of the Berni et al. 2022 paper. Due to the recency of its publication, we hadn’t seen it during the preparation of this manuscript. We were pleased to see similar results in another hemiptera. We have now cited that paper. As for interpreting the varied roles of NBAD and NADA branches in melanization, we do now mention the evident variability and flexibility of this pigmentation system in the summary paragraph (see lines 457-459) but have eschewed incorporating an expanded section of what would be mostly speculative material.